# Validation of depression, anxiety, and stress scales (DASS-21) among Thai nursing students in an online learning environment during the COVID-19 outbreak: A multi-center study

Yuwadee Wittayapun[1,2], Ueamporn Summart[3]*, Panicha Polpanadham[4], Thanyaporn Direksunthorn[5], Raweewan Paokanha[6], Naruk Judabood[7], Muhamad Zulfatul A'la[8]

1 Movement Science and Exercise Research Center-Walailak University (MoveSE-WU), Walailak University, Nakhon Si Thammarat, Thailand, 2 School of Allied Health Sciences, Walailak University, Nakhon Si Thammarat, Thailand, 3 Faculty of Nursing, Roi Et Rajabhat University, Roi Et, Thailand, 4 Faculty of Physical Therapy, Huachiew Chalermprakiet University, Bangkok, Thailand, 5 School of Medicine, Walailak University, Nakhon Si Thammarat, Thailand, 6 Faculty of Nursing, Ratchathani University, Udonthani Campus, Udonthani, Thailand, 7 Faculty of Nursing, Chalermkarnchana University, Srisaket Campus, Srisaket, Thailand, 8 Faculty of Nursing, University of Jember, Jember, Indonesia

* yogiueamporn@gmail.com

**Data Availability Statement:** All relevant data are within the manuscript and its Supporting

## Abstract

The Depression, Anxiety and Stress Scale (DASS-21), an introductory scale used to identify common mental disorders (CMDs) among adults, was validated across cultures in Asian populations; nevertheless, its capacity for screening these disorders may be limited for some specified groups, including nursing students. This study attempted to investigate the psychometric scale's unique features of DASS-21 for Thai nursing students in an online learning environment during the COVID-19 outbreak. A cross-sectional study using the multistage sampling technique recruited 3,705 nursing students from 18 universities located in south and northeast Thailand. The data were gathered using an online web-based survey, and then all respondents were divided into 2 groups (group 1, n = 2,000, group 2, n = 1,705). After using the statistical methods to reduce items, exploratory factor analysis (EFA) using group 1 was performed to investigate the factor structure of the DASS-21. Finally, group 2 used confirmatory factor analysis to verify the modified structure proposed by the EFA and assess the construct validity of the DASS-21. A total of 3,705 Thai nursing students were enrolled. For the factorial construct validity, a three-factor model was initially suggested containing 18 items (DASS-18) spread across 3 components: anxiety (7 items), depression (7 items) and stress (4 items). The internal consistency reliability was acceptable with Cronbach's alpha in the range of 0.73–0.92for either the total or its subscales. For convergent validity, average variance extracted (AVE) showed that all the DASS-18 subscales achieved convergence effect with AVE in the range of 0.50–0.67. The psychometric features of the DASS-18 will support Thai psychologists and researchers to screen CMDs more easily among undergraduate nursing students in tertiary institutions who enrolled in an online learning environment during the COVID-19 outbreak.

information files. We have already generated our minimal data set 2,000 datasets out of the total (3,705) (D1.XLSX).

**Funding:** This study was supported by The Walailak University's Individual Research Grants provided funding for the study (Grant Number WU-IRG-65-015). The funders had no role in study design, data collection and analysis, decision to publish, or preparation of the manuscript.

**Competing interests:** The authors have declared that no competing interests exist.

## Introduction

Depression and anxiety are common mental health disorders (CMDs), leading causes of disability and have gained prominence due to their growing global burden [1]. Individually, these disorders contribute to poor psychological wellbeing, which interferes with learning and inhibits students' academic performance [2]. Early screening for anxiety and depression in primary care and academic settings necessitates an assessment strategy that is rapid and easy to apply and has proven psychometric properties [2].

During the COVID-19 pandemic, undergraduate students also reported their anxiety or stress [3, 4]. According to a network study, 932 nursing students were included. More than one third of these students reported at least moderate symptoms of worry or stress, and almost one half of these students presented at least mild symptoms of depression [5]. Due to a quick shift from face-to-face to an online learning environment during university lock downs, undergraduate nursing and midwifery curricula had trouble adjusting to "remote learning" that relies on the use of electronic technologies and media sources to conduct learning outside of the traditional classroom [6]. Thus, being unable to participate in extensive training such as clinical settings make students feel as though they are passing up a good opportunity to learn these abilities that may have decreased students' mental health [5, 7]. In addition, anxiety and depression may occur more commonly among low experienced apprentices including nursing students [8]. Moreover, this current study also discovered that the levels of anxiety and depression were higher among nursing students than among those students from other disciplines, regarding their probably high risk of infection exposure and fear of communicable diseases [8]. Currently, rare evidence is available concerning the mental health of Thai nursing students encountering an instant psychological response in relation to COVID-19. To provide psychiatric interventions to people experiencing these negative emotional states, early diagnosis of these diseases is essential [2]. Early assessment, using an effective screening instrument (such as a rating scale), provides a rapid indicator of the client's emotional well-being and is helpful for further clinical judgment and early treatment. Likewise, self-reported questionnaires and clinician-rated scales are two commonly used methods to assess CMDs [9].

The Depression, Anxiety, and Stress Scale (DASS) is a common scale frequently used to detect CMDs [10]. Both the DASS-42 and its shortened version, the DASS-21, are frequently used to assess depression, anxiety, and stress among adults, and are considered superior to other psychometric tools to identify these CMDs and screen for psychological abnormalities [11]. Moreover, DASS-21 has several advantages over the original 42-item version (DASS-42), such as fewer items, cleaner factor structure and smaller interfactor correlations [12]. Data analyses among adults using this measure produced consistent results regarding its psychometric properties [2, 7, 12–14]. Findings regarding the DASS-21's factor structure are contradictory, ranging from one to four factors structures [15–17]. Results from a prior study in Asia, conducted among nursing students in Brunei, have validated the DASS-21 used the final model, including a nine-item scale across three components [18]. However, this study encountered limitations because this representative sample (n = 126) was the smallest compared with other studies.

In the Thai context, DASS-21 has been validated across cultures among Asian residents from various projects and research objectives such as assessing the work-related stress and coping strategies among employees in the education and health care sectors [19], so its ability in detecting these mental health problems may be limited for specific groups including undergraduate nursing students. Another study enrolled preclinical medical students to explore psychometric properties of DASS-21, but this study also used this tool as dependent variable and did not report the constructed validity or the Cronbach alpha coefficient [20]. Insufficient data

are available to validate the DASS-21's psychometric properties in specific Thai populations, which could constitute considerable variation concerning sociocultural backgrounds and political differences among groups.

To date, regarding the context of online learning, no research, concerning factor structures and convergent validity of DASS-21, has been conducted among Thai nursing students. Applying the original subscales scoring for all adulthood categories to only young adult age groups (approximately ages 18 to 26 years) without comprehending the instrument's psychometric properties could lead to inaccurate conclusions [21]. This study aimed to examine the psychometric scale-specific features of DASS-21 for Thai nursing students concerning an online learning situation during the COVID-19 outbreak.

## Materials and methods

### Study design and population

This constituted cross-sectional research, obtaining data from one part of the larger multi-center study, aiming to assess the effects of online learning on the prevalence and factors associated with musculoskeletal disorders among Thai, Indonesian, Vietnamese, and Lao faculty members and students during the COVID-19 outbreak. After this study is completed, they recruit some samples and send them the survey of DASS-21. The target population of this study comprised Thai nursing students nationwide undertaken using a multistage sampling technique. Altogether, 96 nursing institutes in Thailand are spread across five regions. Using a simple random sampling technique, two of the five regions, the south and the northeast, were chosen in the first stage. In these areas, 37 nursing institutes are located. Then using a nonproportional stratified sampling technique, 15 nursing institutes were chosen. In addition, three nursing faculties from the Central Region were conveniently sampled, providing a total of 18 institutes. Totally, 5395 students were able to receive participant information sheets online from us. Prior to collecting data, administrators' approval was obtained after a thorough description of the study's objective. Before the survey began, a statement of consent was obtained from all participants by permitting only those who pressed "I consent" to go to the questionnaires. A total of 5136 participants indicated their willingness to join the study. Code numbers were created to protect students' privacy. Finally, nonproportional stratified sampling was used on 4,618 undergraduate students, and 3705 became eligible respondents. Individual student data is only accessible to the authors of this study. These samples were employed to access study participants using an online web-based survey. Recruiting participants and collecting data occurred from April 2022 to June 2022.

### Study instruments

The DASS-21 assesses depression, anxiety, and stress symptoms [10], and is divided into three subscales, each with seven items including depression (DASS 21-D), anxiety (DASS 21-A) and stress (DASS 21-S) (DASS 21S). The translation of this tool from English to Thai was carried out during the cross-culture translation procedure [19]. Each item is graded on a four-point Likert scale ranging from 0 ("did not apply to me at all") to 3 ("applied to me a lot"). Because the DASS-21 is a shortened version of the DASS (42 items), the final score of each scale was multiplied by two before being compared with the original DASS scale. Higher scores and response values reflect greater levels of the condition being evaluated. In this study we used the Thai version of the DASS-21 with the original author's permission [19]. The Cronbach's alpha coefficients of depression, anxiety, and stress for The Thai version are 0.82, 0.78, and 0.69, respectively [19].

The Visual Analog Scale to Evaluate Fatigue Severity (VAS-F) [22] has 18 components all related to one's perception of exhaustion. Each question asks respondents to place an "X" along a VAS line that runs between two extremes, such as "not at all fatigued" and "very tired," to identify what they are feeling right now. The score goes from 0 to 100 and is recorded using a vertical line of 10 cm. The line from "No fatigue" to the subject's stated point indicating their level of fatigue, was measured to obtain the score; the higher the VAFS score, the greater the level of fatigue [23]. The Cronbach's alpha for the fatigue subscale was 0.91 and the value of energy subscale was 0.94, respectively [22]. In addition, questions about general information of the participants, i.e., gender, age, study year, online learning, were included.

## Sample size calculation

The sample size was calculated using the formula "sample size = number of items X number of participants," which is an extensively used formula in survey development research. We estimated the minimum sample size based on one item to ten participants [24]. Therefore, the minimum acceptable sample size, based on 21 items of DASS-21, was 210 respondents. However, our research enrolled 3,705 nursing students from 18 universities mainly located in south and northeast Thailand. Hence, larger sample size could provide more meaningful factor loadings and yield more generalizable results. The inclusion criteria for the participants were age at least 17 years, a nursing student at the institute during the study period more than six months and engaged in the online learning. Individuals who do not fill the administered questionnaire or submit an incompletely filled questionnaire such as responding to only a part of general information in the Thai Version of DASS-21, and nursing students with existing CMDs were excluded from this study. To avoid model overfitting, the exploratory (EFA) and confirmatory factor analyses (CFA) were organized on a random split of the total 3705 subjects in two group samples (group 1, n = 2000, and group 2, n = 1705).

## Statistical analysis

All statistical analysis in this study was conducted using IBM SPSS and AMOS version 20. Descriptive statistics with means and standard deviation for continuous variables and counts and percentages for categorical data were used to describe the participant's demographic characteristics.

To investigate the number of components in the EFA for the DASS-21 measuring model, parallel analysis (based on principle component analysis) was undertaken using sample group 1. Then the structure of factors was investigated using principal axis factoring with varimax rotation. Factor loadings less than 0.5 were suppressed, and item cross loadings more than 0.2 were removed one at a time. Furthermore, factor loadings were used to calculate average variance extracted (AVE) and composite reliability (CR). Regarding the findings of the principal axis factoring, a CFA was applied to the remaining held-out participants. The measurement model was fitted using an unweighted least square estimate CFA, and model fit was evaluated using the cumulative fit index (CFI), adjusted goodness of fit index (AGFI), root-mean-square error of approximation (RMSEA), and Tucker-Lewis's index (TLI) [19, 25, 26]. Likewise, Root Mean Square Error of Approximation (RMSEA) with a p-value less than 0.08 was considered to indicate a good model fit, so it was reported and used in this investigation for the sake of convention. Along with the CFA, the Kaiser-Meyer-Olkin (KMO) measure of sampling adequacy and the Bartlett's test of sphericity were developed to provide additional construct validity evidence [27].

CR and Cronbach's alpha were used to assess reliability. CR is acceptable when the values for the three subscales are greater than 0.6 [28]. Cronbach's alpha was used to assess internal

consistency reliability, and Cronbach's alpha above 0.7 for all the subscales was considered to be an acceptable reliability [28]. In addition, the relationship between each of the DASS items and its own DASS subscale with that item removed is known as the corrected item-total correlations of the three subscales.

We investigated the convergent validity of DASS-18 using the AVE. To indicate convergent validity, the AVE must be equal to or greater than 0.50, indicating that the construct's variance accounts for more than 50% of its variation [26]. Furthermore, the discriminant validity determined whether the three indicators of depression, anxiety and stress domains were distinct factors from one another. Pearson's correlation (r) lower than 0.85 among variables verified their discriminant validity [26] Pearson correlations were calculated to investigate the intercorrelations matrix, the temporal stability of DASS-18 subscale scores and the relationship between DASS-18 and VAS-F.

## Ethics considerations

This study was examined and authorized by Walailak University's institutional review board (Ref. No. WUEC-22-007-01) and the Center for Ethics in Human Research, Khon Kaen University (Ref. No. HE652094).

## Results

### General information of the participants

The details of participant characteristics are described in Table 1. Most completing the questionnaire were females (94.2%) with a mean age of 20 (SD = 1.26) years. Almost two thirds of these participants (67.2%) were experiencing some semesters of online learning at the time of collecting data.

### Exploratory factor analysis

After randomizing the 2000 participants for EFA, firstly, parallel analysis of the matrix indicated that a three-factor solution could be extracted. Secondly, the rotational factor loading matrix was statistically significant. Three factors having eigenvalues over one were created by

**Table 1. Demographic characteristics of the participant.** (n = 3705).

| Baseline characteristic | Number | Percent |
|---|---|---|
| Sex | | |
| Male | 215 | 5.8 |
| Female | 3490 | 94.2 |
| Age (year) | | |
| 18–19 | 1137 | 30.7 |
| $\geq$20 | 2568 | 69.3 |
| Mean (SD) | 20.0 (1.26) | |
| Study year | | |
| 1st | 1313 | 35.4 |
| 2nd | 1011 | 27.3 |
| 3rd | 767 | 20.7 |
| 4th | 614 | 16.6 |
| Online learning | | |
| 100% | 1213 | 32.8 |
| Some semester/subject | 2492 | 67.2 |

**Table 2. Exploratory factor analysis of the DASS-18.**

| Code | Item | Factor loading [a] | | |
|---|---|---|---|---|
| | | DASS-18-D | DASS-18-A | DASS-18-S |
| D3 | I could not seem to experience any positive feeling at all. | 0.756 | | |
| D5 | I found it difficult to work up the initiative to do things. | 0.758 | | |
| D10 | I felt that I had nothing to look forward to. | 0.790 | | |
| D13 | I felt downhearted and blue. | 0.620 | | |
| D16 | I was unable to become enthusiastic about anything. | 0.607 | | |
| D17 | I felt I wasn't worth much as a person. | 0.708 | | |
| D21 | I felt that life was meaningless. | 0.711 | | |
| A2 | I was aware of dryness of my mouth. | | 0.853 | |
| A4 | I experienced breathing difficulty. | | 0.863 | |
| A7 | I experienced trembling. | | 0.843 | |
| A9 | I was worried about situations in which I might panic and make a fool of myself. | | 0.856 | |
| A15 | I felt I was close to panic. | | 0.849 | |
| A19 | I was aware of the action of my heart in the absence of physical exertion. | | 0.597 | |
| A20 | I felt scared without any good reason. | | 0.852 | |
| S1 | I found it hard to wind down. | | | 0.692 |
| S6 | I tended to over-react to situations. | | | 0.798 |
| S14 | I was intolerant of anything that kept me from getting on with what I was doing. | | | 0.729 |
| S18 | I felt that I was rather touchy. | | | 0.745 |
| | KMO | 0.956 | | |
| | Cronbach Alpha | 0.825 | 0.968 | 0.900 |
| | Total | 1.278 | | |
| | Cumulative % | 71.311 | | |

[a] Extraction Method: Principal Component Analysis.

the initial analyses of the Group 1 sample. To ascertain the factorial structure of DASS-21 and the underlying dimensions comprising its 21 items. The initial analysis revealed a three-factor structure that explained 69.31% of the original data's variance. Three items (S8, S11, S12) from the stress scale were found to be loading on multiple factors; therefore, these items were removed from this analysis. The three factors resembled the original structure (9) with a reduced factor in stress component; however, the three-factor component (eigenvalues = 9.82; 1.74; and 1.23) was revealed by the scree plot and the eigenvalues higher than one requirement, and this model accounted for 71.31% of the variance. The result of the KMO test was 0.965 ($\chi2$ = 30932; p<0.001), showing that the model was highly adequate. The factor loadings for each DASS- 18 item are shown in Table 2, with factor loadings >0.50 indicating acceptable loading (Table 2).

## Confirmatory factor analysis

Three items from the stress scale were eliminated (the remaining 18 items of DASS, thus DASS-18). The DASS-18 measuring model, which included 18 items distributed across three components including DASS-18-A (7 items), DASS-18-D (7 items) and DASS-18-S (4 items) was fitted using an unweighted least square CFA. Based on the five specified fit criteria, the model demonstrated an acceptable fit to the data (CMIN/df = 3.082; p = 0.001; CFI = 0.98; RMSEA = 0.032; GFI = 0.98 and NFI = 0.99. The effect of the large sample size may have prevented the chi-square tests from providing acceptable assessments of model fit, whereas other

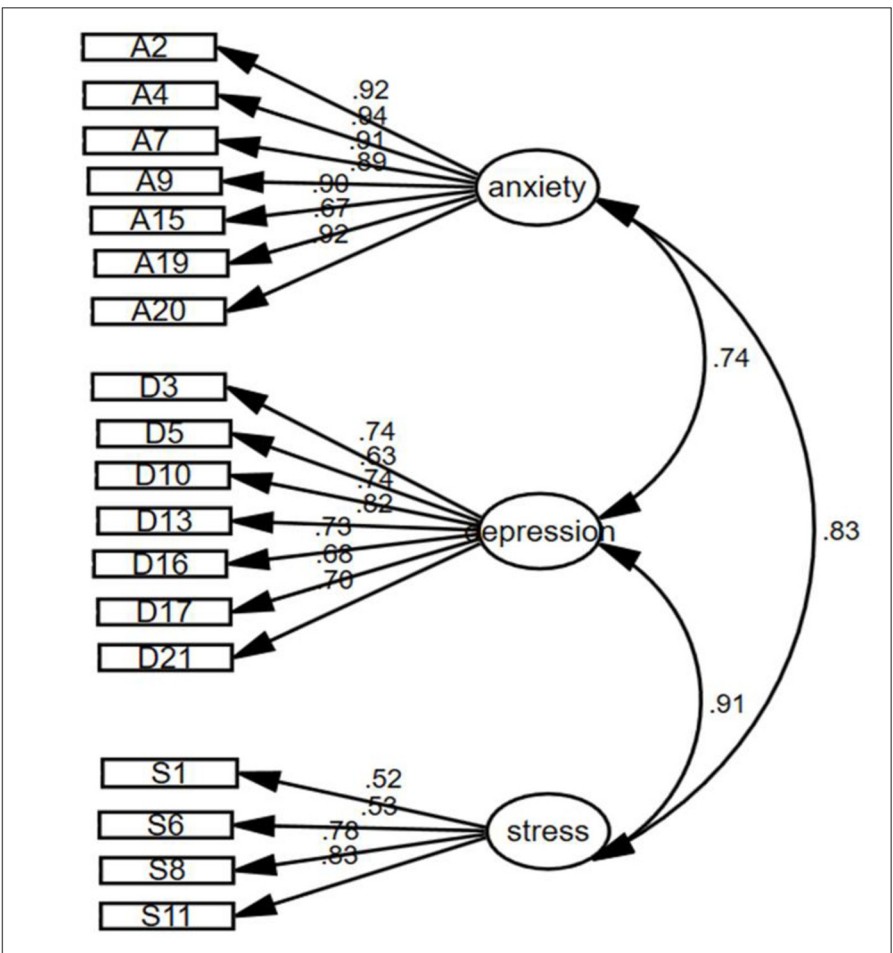

**Fig 1. Structure model of the DASS-18 with standardized path coefficient.**

indices indicated that these models remained well-fitted for the data. In addition, except for each factor-constraint item, so that no significant test could be archived, all model items were significantly loaded a long with their concurrent factors (all p-values <0.05) (Fig 1).

In all cases, the Pearson's correlation coefficients between DASS-18-D, DASS-18-A and DASS-18-S presented moderate to remarkably elevated levels (0.52 to 0.92) indicating that these scales were moderately to highly discriminatory.

## Convergent validity

AVE calculations showed all the DASS-18 subscales achieved a convergence effect (with the AVE of depression = 0.504; the AVE of anxiety = 0.674 and the AVE of stress = 0.551).

## Discriminant validity

The magnitude of the correlations among depression, anxiety and stress domains determined the discriminant validity of the variables (Fig 1). The variables showed correlations (r) lower than 0.85 except the correlation between depression and stress domains (r = 0.91).

## Association of the DASS-18 scores among demographic characteristic and VAS-F

The total scale of DASS-18 showed a statistically significant positive and fair to moderate association with the VAS-F total score, sex and online learning. Additionally, the DASS-18 total score showed a moderately positive significance correlated to VAS-F in the anticipated direction, confirming the association between higher levels of fatigue and higher levels of depression, anxiety, and stress (Table 3).

## Reliability analysis

The CR of the three domains of DASS-18 ranging from 0.830 to 0.935 indicated evidence of acceptable reliability. Regarding Cronbach alpha values of 0.92 for the overall scale, 0.87 for depression, 0.79 for anxiety and 0.73 for stress, the DASS-18 exhibited adequate internal consistency reliability. Similarly, the internal consistency of this scale was good, as evidenced by the item-rest correlations for all three subscales being better than 0.3 and the corrected item-rest correlation for the entire scale ranging from 0.53 to 0.91 (Table 4).

Product moment intercorrelations matrix values were determined between the three domains of DASS-18 and VAS-F. These intercorrelations values were found to be moderately strong the subscales of depression and anxiety showed the strongest intercorrelation among the three (r = 0.735), which was also statistically significant. These results could imply that the stress domain of DASS-18 moderately and positively correlated to VAS-F (r = 0.445) (Table 5).

## Discussion

The purpose of this study was to evaluate the psychometric properties of DASS-21 among Thai nursing students experiencing online classes during the COVID-19 outbreak. For the results of the factorial construct validity of the DASS-18, a three-factor model showed satisfactory conformity to the psychometric construct of the DASS-21 original version [10], and these results support the fact that the DASS-18 instrument for this cohort contained 18 items spread across three components as follows: anxiety (seven items), depression (seven items) and stress (four items). The three factors were comparable to the structure found by prior studies exploring the psychometric features and generalization of the DASS-21 for use in Asian nations [19]. This investigation showed that the DASS-18 is a promising and psychometrically sound tool, ideally suited for determining the frequency and intensity of symptoms associated with negative affective states for these participants. Furthermore, the two-week temporal stability was good for all DASS-18 scale scores; in particular, the DASS-18 stress subscale showed the highest correlation values across time and they had a great internal consistency reliability, agreeing with our hypotheses. The consequences of reducing three items from the stress scale are the reasons for the lower Cronbach's alpha coefficient of this scale than that of Lovibond and Lovibond's original version [10]. These differences might have resulted from the DASS-18 having

**Table 3. Association of the DASS-18 scores with demographic characteristics of the Thai nursing student sample.**

| Variable | Total scale of DASS-18 | p-value |
|---|---|---|
| Age | 0.151 | 0.763 |
| Sex | 0.103 | <0.001 |
| Online learning | 0.451 | <0.001 |
| VAS-F | 0.461 | <0/001 |

**Table 4. Reliability analysis of the DASS-18.**

| Subscale | Composite reliability | Cronbach's alpha | Item test correlation |
|---|---|---|---|
| Depression | 0.876 | 0.87 | 0.65–0.77 |
| Anxiety | 0.935 | 0.79 | 0.66–0.91 |
| Stress | 0.830 | 0.73 | 0.53–0.63 |
| Overall | - | 0.92 | 0.57–0.81 |

fewer items because the quantity of items creates an impact on how Cronbach's alpha is calculated [19].

According to our results, only minimal changes were observed between the original DASS-stress (seven items) and our DASS-stress (four items) scales. These disparities might be explained by how diverse culture's view perception of some items that could be interpreted as besides the stress context cultural factors and the response of the participants may influence how individuals understand item of the DASS-stress scales, but not on the DASS-depression and DASS-anxiety scale as we found no significant cultural problem with these two scales and no concerns were noted regarding the EFA findings as demonstrated by the statistical results of this study. Therefore, the DASS-18 factor structure clearly demonstrated three factors, as in the original DASS-21 scale [10]. Likewise, one study reported that no invariances were discovered in their multi-group analysis across the six countries [19]. In addition, findings from this previous study on the correlations of the three subscales with those of other psychiatric instruments measuring similar constructs offered support for the validity of the DASS-18 subscales and were generally favorable [19]. Moreover, the depression and anxiety subscales of the DASS-18 exhibited specific relationships with the relevant measures of these disorders, indicating that using these constructs was appropriate.

Cronbach's alpha coefficient from our study revealed that total DASS-18 scores and its subscales exhibited good internal consistency. This coefficient ranged from good to excellent in prior studies comprising both nonclinical and clinical adult samples [2, 9, 10, 19, 21, 29–31]. Hence, the data collectively proved that the DASS-18 demonstrated strong internal consistency across a variety of demographics and languages. Moreover, the results of the item analysis indicated that the items in each scale had good discrimination indices (corrected item-total correction). These indices suggested that the DASS-18 Thai version items would be effective at distinguishing between high and low scorers on this scale. Related research has also revealed that this assessment tool provides a good item discrimination index [13, 18].

The relationships between demographic characteristics and DASS-18 scale scores were also investigated. This study found a weak positive statistically significant relationship between DASS-18 and sex. Despite the concerns about future endurance and competency aspect, female participants expressed more depression, anxiety, and stress than males. This may be

**Table 5. Intercorrelations matrix between domains of DASS-18 and VAS-F.**

| Scales | DASS-D | DASS-A | DASS-S | VAS-F |
|---|---|---|---|---|
| DASS-D | 1.00 | 0.735* | 0.737* | 0.417* |
| DASS-A | 0.735* | 1.00 | 0.708* | 0.396* |
| DASS-S | 0.737* | .708* | 1.00 | 0.445* |
| VAS-F | 0.417* | 0.396* | 0.445* | 1.00 |

* All correlations are significant at p <0.001 level (2-tailed)

because female nursing students usually have commitments outside of the classroom, such as taking care of their family members and performing chores [7, 32]. Our results indicated that online learning moderately, positively correlated to the total DASS-18 score because high standards for performance, learning habits, and training may negative impact students' mental health [7]. Similarly, clinical courses in nursing programs call for specialized cognitive, emotional, and psychomotor abilities typically following specialized theoretical courses. Because of being unable to take part in clinical settings, these students felt as though they were missing out on a great opportunity to acquire these abilities [6, 20]. Thus, these students may have felt unprepared for learning in a clinical setting due to the extremely brief on campus learning time before lockdown, and the pandemic made it more difficult for nursing students to advance in their practical training [6]. When lockdowns ended, nursing students had greater opportunity to contract an illness by themselves or face patients with COVID-19 experiencing significant effects [5].

The internal consistency of the DASS-18 was adequate and consistent with the related Asian research [7, 19]. The Thai version's convergent validity is supported by favorable correlations with the Beck Depression Inventory (BDI), the Beck Anxiety Inventory (BAI); correlations in this direction were anticipated to measure the same construct [19]. These results demonstrated the validity and reliability of the Thai DASS-18 version as a tool for measuring negative emotional states. This indicated that this scale could prove beneficial for screening CMDs among clinical undergraduate students including nursing students.

The convergent validity of the DASS-18 was examined using the AVE calculation of all three subscales. The results revealed that all sub-scales' AVE were greater than 0.50, corresponding to the convergent validity acceptance criteria. These findings were also compatible with the findings of a study that aimed to validate the DASS-21 among Vietnam students in an e-learning environment [7]. Regarding the discriminant validity, factors of this stress subscale also highly correlated to each other, which were higher than the values suggested by Hu and Bentler [26]. These higher correlations indicated significant overlapping in the content of the DASS-18 scales, indicating a general construct, such as affective distress. One related study also reported a higher correlation among these subscales [13].

When comparing depression, anxiety and stress scales, anxiety items had higher factor loadings, eigenvalues, and percentage of variation than the other domains. Depression and anxiety continued to have the highest inter-correlations, with a value of 0.708 indicating significant overlap between the two domains. Despite the overlap between domains, they could still be separated. An extraordinarily strong and positive association was noted between these domains. The Thai nursing students' symptoms of stress, anxiety and depression were all positively connected, according to these positive correlation values. The DASS-18's correlation coefficients showed beneficial correlations between the two instruments in this regard. Likewise, these coefficients also showed that the subjects' anxiety and depression were present at the same time. The DASS-18 was thus shown to measure depression and anxiety among the responders in a simultaneous and unique manner. These findings were in line with studies in other countries [21, 29].

## Strengths and limitations

The strength of this study is the fact that the structure and psychometric features of the DASS-21 were examined for the first time in a large sample of undergraduate nursing students in Thailand. Because the participant-to-questionnaire-item ratio was satisfactory, the prerequisites for component analysis were met, and bias resulting from the number of observations was reduced.

Our study encountered limitations regarding the nursing students comprising our subjects. They could not accurately be generalized to other nonclinical undergraduate students due to their diverse qualities because they may be privileged in terms of socioeconomic status, freedom, and health. The study was cross-sectional; hence, the data were unable to show test-retest reliability over time. The study was limited in terms of criterion validity because we did not test any other parameters besides VAS-F.

## Conclusion

The study 's main findings demonstrate that the DASS-18 is a valid instrument for detecting CMDs among Thai nursing students enrolled in online courses during the COVID-19 outbreak. The three-factor structure with 18 items proposed in the initial study was supported by the findings. Therefore, the availability of the DASS-18's psychometric features will enhance performance of Thai psychologists and researchers in effectively screening the population of undergraduate nursing students for CMDs at tertiary institutions.

## Supporting information

**S1 Data.**
(XLSX)

## Acknowledgments

The authors thank all nursing institutions for supporting students' data collection. The research collaborators at all involved nursing institutes are thanked by the authors for helping with sample recruitment and data gathering. All nursing students joining this study are gratefully acknowledged.

## Author Contributions

**Conceptualization:** Yuwadee Wittayapun.

**Data curation:** Yuwadee Wittayapun.

**Formal analysis:** Yuwadee Wittayapun, Ueamporn Summart.

**Funding acquisition:** Yuwadee Wittayapun.

**Investigation:** Yuwadee Wittayapun, Ueamporn Summart, Panicha Polpanadham, Raweewan Paokanha, Naruk Judabood, Muhamad Zulfatul A'la.

**Methodology:** Yuwadee Wittayapun, Ueamporn Summart.

**Visualization:** Yuwadee Wittayapun.

**Writing – original draft:** Yuwadee Wittayapun, Ueamporn Summart, Raweewan Paokanha, Naruk Judabood.

**Writing – review & editing:** Yuwadee Wittayapun, Ueamporn Summart, Panicha Polpanadham, Thanyaporn Direksunthorn, Raweewan Paokanha, Muhamad Zulfatul A'la.

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
