## [Decision Letter · Decision Letter 0]

10 Apr 2023

PONE-D-22-31148Ref. KPB.B 1900/225/2565 11 December, 2022Validation of the depression, anxiety, and stress scales (DASS-21): among Thai nursing students in the online learning environment during the COVID-19 outbreak: A multi-center study.PLOS ONE

Dear Dr. summart,

Thank you for submitting your manuscript to PLOS ONE. After careful consideration, we feel that it has merit but does not fully meet PLOS ONE’s publication criteria as it currently stands. Therefore, we invite you to submit a revised version of the manuscript that addresses the points raised during the review process.

We look forward to receiving your revised manuscript.

Kind regards,

Omar M Khraisat, Associate Professor

Academic Editor

PLOS ONE

2. During your revisions, please note that a simple title correction is required: Please remove the text "Ref. KPB.B 1900/225/2565 11 December, 2022" from the title on the online submission information.

4. Thank you for stating the following in the Acknowledgments/ Funding Section of your manuscript:

“The Walailak University's Individual Research Grants provided funding for the study (Grant Number WU-IRG-65-015).”

6. We note that you have indicated that data from this study are available upon request. PLOS only allows data to be available upon request if there are legal or ethical restrictions on sharing data publicly. For more information on unacceptable data access restrictions, please see http://journals.plos.org/plosone/s/data-availability#loc-unacceptable-data-access-restrictions.

Reviewers' comments:

Reviewer's Responses to Questions

**Comments to the Author**

1. Is the manuscript technically sound, and do the data support the conclusions?

Reviewer #1: Partly

2. Has the statistical analysis been performed appropriately and rigorously? 

Reviewer #1: No

3. Have the authors made all data underlying the findings in their manuscript fully available?

Reviewer #1: No

4. Is the manuscript presented in an intelligible fashion and written in standard English?

Reviewer #1: No

5. Review Comments to the Author

Reviewer #1: 1. Under study design & Population: It is not clear how they recruit the participants in their study since the sample was taken from a larger multi-center survey, and how they randomize them.

2.Do the survey was part of the larger study, or after the larger study completed they recruit the sample and send them the survey of DASS-21

2. "To interpret these results is comparable to the original version of DASS-41 that calculated from the total scores for each sub-scale multiplying by two". This statement is not clear. & why they have used the 21 items scale since the original was 41 items?

3. Reference number 24 is retracted.

4. Based on what you have enrolled this number of participant (3705)? and you have mentioned that you need 5 – 10 participants for each item.

5. The exclusion criteria are not clear in the manuscript.

6. Fatigue (VAS-F) is different construct than anxiety and depression. So how it was used to test the convergent validity of the DASS-18.

7. It is not clear how randomization of the participants was done?

8. It was mentioned that the parellel analysis revealed a single factor, then a 3 factor solution was created. how this was run?

9. What was the factor loading for the deleted items and in which factors they loading(the rule is to retain items if they had a factor loading of 0.4 or more).

10. Did the total variance changed after you remove the three factors?

11. The title of Table 2 need to be changed to "The title of the table should be Exploratory Factor Analysis of the DASS-18"

12. in page 11 under the tittle Confirmatory factor analysis, the term "subscales" is used instead of "Items".. there is a big difference between both.

13. What was Composite Reliability (CR) of the three factors as an evidence of reliability.

14. What was the Average Variance Extracted (AVE) for the scales ? (this is considered a measure of convergent validity)

15. What is the relation of gender to convergent validity? Gender is not a scale. how you consider is an evidence of convergent validity?

16. Table 3 has a lot of errors & empty cells. besides, this is a correlation table not a Convergent and discriminant validity for the subscale of the DASS-18.

17. How did you measure the variable "Online Learning". it was mentioned in table 3.

18. some non scientific terms are used such as "directly linked".. "values were discovered to exist"." comparable discoveries"

19. I am not sure why table 5 was reported. it is a correlation table. they have already run the factor analysis.

20. How the temporal stability of the tool was measure. " the term is mentioned in page 13 under "Discussion".

21. There are some typing and grammatical errors in the manuscript.

22. How did the authors know that female had more depression? We can’t know that by simple running correlation. You need to do a T-test

23. The authors have mentioned in the discussion that "no effect on both the DASS-depression and DASS-anxiety scales as we found no significant cultural problems with those two scales and there was no concern with the EFA findings as demonstrated by the statistical results". I think this couldn’t attributed to cultural factors only. This depends on the responses of the participants

24. The authors have mentioned that "The Thai version's convergent validity is supported by favorable correlations with VAS-F dimensions; correlations in this direction were anticipated which were expected to measure the same construct" Actually they are measuring different constructs, so they came up with this conclusion?

25. the authors have reported that there is an evidence of concurrent validity but it is not clear how it was measured (278 anxiety and depression are used in the same tool, so it can't be considered a concurrent validity.

6. PLOS authors have the option to publish the peer review history of their article (what does this mean?). If published, this will include your full peer review and any attached files.

Reviewer #1: No

---

## [Author Response · Author response to Decision Letter 0]

1 May 2023

For the reviewser' scomments, I am very appreciate for all comments because I have some error before you kindly comment our manuscript. Thanks for your good comments for me.

---

## [Decision Letter · Decision Letter 1]

8 Jun 2023

PONE-D-22-31148R1Validation of the depression, anxiety, and stress scales (DASS-21) among Thai nursing students in the online learning environment during the COVID-19 outbreak: A multi-center study.PLOS ONE

Dear Dr. summart,

Thank you for submitting your manuscript to PLOS ONE. After careful consideration, we feel that it has merit but does not fully meet PLOS ONE’s publication criteria as it currently stands. Therefore, we invite you to submit a revised version of the manuscript that addresses the points raised during the review process.

We look forward to receiving your revised manuscript.

Kind regards,

Omar M Khraisat, Associate Professor

Academic Editor

PLOS ONE

Journal Requirements:

Reviewers' comments:

Reviewer's Responses to Questions

**Comments to the Author**

1. If the authors have adequately addressed your comments raised in a previous round of review and you feel that this manuscript is now acceptable for publication, you may indicate that here to bypass the “Comments to the Author” section, enter your conflict of interest statement in the “Confidential to Editor” section, and submit your "Accept" recommendation.

Reviewer #2: All comments have been addressed

Reviewer #3: All comments have been addressed

2. Is the manuscript technically sound, and do the data support the conclusions?

Reviewer #2: Yes

Reviewer #3: Yes

3. Has the statistical analysis been performed appropriately and rigorously? 

Reviewer #2: Yes

Reviewer #3: Yes

4. Have the authors made all data underlying the findings in their manuscript fully available?

Reviewer #2: Yes

Reviewer #3: Yes

5. Is the manuscript presented in an intelligible fashion and written in standard English?

Reviewer #2: Yes

Reviewer #3: Yes

6. Review Comments to the Author

Reviewer #2: Thank you for the opportunity to review this manuscript. The article reports the validation of the Depression, Anxiety, and Stress Scales (DASS-21) among Thai nursing students in the online learning environment during the COVID-19 outbreak. The topic holds significant importance for the nursing profession, considering the global impact of the COVID-19 pandemic. I particularly appreciate the inclusion of universities from various regions of Thailand in their study. I commend the authors for undertaking this research.

Please see my comments below for strengthening the manuscript:

Introduction: The introduction section could be written with more clarity and focus on the research gap. Some statements are too generic and require context and clarification of terms. I have detailed them below:

• The reference to Lee et al (2019) was used to support the statement about low performance at work or school, disruptions in social life, and even suicide, but the reference is a study evaluating the Depression Anxiety Stress Scales, which is not directly relevant. It would be helpful to provide more specific and relevant references to support these claims.

• The statement about "more than a third of nursing students" and almost 'one half of these students' would be unclear without the total number of nursing students in the context of the study. Clarifying the total number of nursing students would provide better context.

• It would be beneficial to provide a clear definition of 'remote learning' or 'online learning,' whichever terms the authors choose, to establish the context. Additionally, the terms "high intentions," "extensive training and learning processes" need further clarification and explanation.

• The article mentions a research article explaining the high risk of infection exposure and fear of transmitting diseases among students, but it is not clear how this relates to the study's context of 'remote learning'. The connection between remote learning and these factors needs to be clearly established.

• The sentence in line 42-43 requires evidence to support the claim made. It would be beneficial to provide supporting references or data to strengthen this statement.

• There should be a brief explanation of DASS, its psychometric properties, and the difference between DASS-21 and the original DASS scale. Justifying the use of DASS-21 in this study would enhance the manuscript.

• It is important to define the young adult age group being referred to in the study. Clarifying the specific age range would eliminate ambiguity and improve the precision of the research.

• Considering the Thai context during the COVID-19 outbreak/pandemic, it would be valuable to include relevant information about the situation in Thailand, particularly during the data collection phase from April to June 2022. Providing contextual information would enhance the understanding of the study's findings.

Study instruments: I was unsure how the process of translating the original version into Thai was conducted, as the reference was to a DASS Brunei version. It would be helpful to provide information on who conducted the translation, whether there was any back-translation, and the measures taken to ensure there was no loss of meaning in the translation and to ensure consistency and validity. Additionally, information on the reliability and validity of the selected instruments, including the VAS-F scale, should be provided. Was there any pilot testing of the translated version?

Sample size calculation: Information on the sample could have been better. Who were these nursing students? What kind of remote learning did they engage in, how many hours, and what courses were included? How were they recruited? It would also be helpful to clarify whether students with existing CMDs (Common Mental Disorders) were excluded or not.

Line 130 - (group 1, n= 2,000, and group 2, n= 1,706). The total should be 3,706. Please verify this.

Statistical analysis: Can you include the statistical software used? I wasn't clear which items were removed and how the study ended up using DASS-18. It was only explained later on under the subheading "Confirmatory factor analysis." It would be helpful to signpost the reader by mentioning that this would be discussed later on.

The results and discussion sections are generally well-written. However, the statement in Line 274-276 should elaborate more on cultural factors and draw on relevant literature to support the discussion. Additionally, in relation to this statement, I wasn't sure what was meant by "no significant cultural problems." It would be beneficial to provide further clarification or explanation to ensure understanding.

Reviewer #3: (No Response)

7. PLOS authors have the option to publish the peer review history of their article (what does this mean?). If published, this will include your full peer review and any attached files.

Reviewer #2: No

Reviewer #3: **Yes: **Emmanuel Z. Chona

---

## [Author Response · Author response to Decision Letter 1]

13 Jun 2023

Thank you for all of your comment to improve our manuscript. We attemp to edit and clearify all of the statements that you recommends however our revised manuscript is not completely edited. Please accept our apologize. We remain accetable for all of your suggestion.

---

## [Decision Letter · Decision Letter 2]

19 Jun 2023

Validation of the depression, anxiety, and stress scales (DASS-21) among Thai nursing students in the online learning environment during the COVID-19 outbreak: A multi-center study.

PONE-D-22-31148R2

Dear Dr.,

We’re pleased to inform you that your manuscript has been judged scientifically suitable for publication and will be formally accepted for publication once it meets all outstanding technical requirements.

Kind regards,

Omar M Khraisat, Associate Professor

Academic Editor

PLOS ONE

Additional Editor Comments (optional):

Reviewers' comments:

Reviewer's Responses to Questions

**Comments to the Author**

1. If the authors have adequately addressed your comments raised in a previous round of review and you feel that this manuscript is now acceptable for publication, you may indicate that here to bypass the “Comments to the Author” section, enter your conflict of interest statement in the “Confidential to Editor” section, and submit your "Accept" recommendation.

Reviewer #2: All comments have been addressed

2. Is the manuscript technically sound, and do the data support the conclusions?

Reviewer #2: No

3. Has the statistical analysis been performed appropriately and rigorously? 

Reviewer #2: Yes

4. Have the authors made all data underlying the findings in their manuscript fully available?

Reviewer #2: Yes

5. Is the manuscript presented in an intelligible fashion and written in standard English?

Reviewer #2: Yes

6. Review Comments to the Author

Reviewer #2: I am pleased with the comprehensive revisions made in response to the comments. I would like to note that there is a discrepancy in line 144-5, where the total number is stated as 3,706 instead of 3,705 as mentioned elsewhere in the manuscript. Please double-check this inconsistency.

7. PLOS authors have the option to publish the peer review history of their article (what does this mean?). If published, this will include your full peer review and any attached files.

Reviewer #2: No

---

## [Editor Report · Acceptance letter]

22 Jun 2023

PONE-D-22-31148R2 

Validation of depression, anxiety, and stress scales (DASS-21) among Thai nursing students in an online learning environment during the COVID-19 outbreak: a multi-center study. 

Dear Dr. Summart:

I'm pleased to inform you that your manuscript has been deemed suitable for publication in PLOS ONE. Congratulations! Your manuscript is now with our production department. 

Kind regards, 

on behalf of

Dr. Omar M Khraisat 

Academic Editor

PLOS ONE